# Ferritinophagy via NCOA4 is required for erythropoiesis and is regulated by iron dependent HERC2-mediated proteolysis

Joseph D Mancias[1,2,5†], Laura Pontano Vaites[1†], Sahar Nissim[3,4,6], Douglas E Biancur[2], Andrew J Kim[3], Xiaoxu Wang[2], Yu Liu[1], Wolfram Goessling[3,4,6,7,8], Alec C Kimmelman[2]*, J Wade Harper[1]*

[1]Department of Cell Biology, Harvard Medical School, Boston, United States; [2]Division of Genomic Stability and DNA Repair, Department of Radiation Oncology, Dana-Farber Cancer Institute, Boston, United States; [3]Gastroenterology Division, Brigham and Women's Hospital, Harvard Medical School, Boston, United States; [4]Genetics Division, Brigham and Women's Hospital, Boston, United States; [5]Department of Radiation Oncology, Beth Israel Deaconess Medical Center, Harvard Medical School, Boston, United States; [6]Dana-Farber Cancer Institute, Boston, United States; [7]Harvard Stem Cell Institute, Cambridge, United States; [8]Broad Institute of MIT and Harvard, Cambridge, United States

**Abstract** NCOA4 is a selective cargo receptor for the autophagic turnover of ferritin, a process critical for regulation of intracellular iron bioavailability. However, how ferritinophagy flux is controlled and the roles of NCOA4 in iron-dependent processes are poorly understood. Through analysis of the NCOA4-FTH1 interaction, we demonstrate that direct association via a key surface arginine in FTH1 and a C-terminal element in NCOA4 is required for delivery of ferritin to the lysosome via autophagosomes. Moreover, NCOA4 abundance is under dual control via autophagy and the ubiquitin proteasome system. Ubiquitin-dependent NCOA4 turnover is promoted by excess iron and involves an iron-dependent interaction between NCOA4 and the HERC2 ubiquitin ligase. In zebrafish and cultured cells, NCOA4 plays an essential role in erythroid differentiation. This work reveals the molecular nature of the NCOA4-ferritin complex and explains how intracellular iron levels modulate NCOA4-mediated ferritinophagy in cells and in an iron-dependent physiological setting.

*For correspondence:
Alec_Kimmelman@dfci.harvard.edu (ACK); wade_harper@hms.harvard.edu (JWH)

†These authors contributed equally to this work

Competing interests:
See page 17

## Introduction

Iron is essential for the survival of nearly all organisms as it serves as a cofactor for a host of biochemical processes including oxygen storage, oxidative phosphorylation, and enzymatic reactions required for cellular proliferation (*Pantopoulos et al., 2012*). However, the levels of free iron in a cell must be tightly controlled to avoid the generation of reactive oxygen species (ROS) via the Fenton reaction (*Dixon and Stockwell, 2014*). As such, iron metabolism is a tightly regulated process controlled by a network of iron-dependent proteins (*Pantopoulos et al., 2012*). The cell has evolved mechanisms whereby iron can be sequestered and released from protein complexes in response to changing iron levels (*Anderson et al., 2012*). One such protein is ferritin, which forms a complex of 24 subunits (consisting of a mixture of ferritin heavy and light chains, FTH1 and FTL, respectively) capable of storing up to 4500 iron atoms (*Arosio et al., 2009*). When iron levels in the cell are low, ferritin is degraded allowing the release of iron for use by the cell. Ferritin is degraded in the lysosome and recent evidence implicated autophagy, a conserved catabolic cellular pathway, in the 'ferritinophagy'

**eLife digest** The cells of nearly all organisms need iron as this metal plays an important role in a wide range of biological processes. However, iron can also trigger the formation of harmful molecules that can damage cells. It is therefore crucial that the amount of iron in cells is tightly controlled and that any extra iron is safely stored away. Most of the iron in the body is stored within a protein called ferritin, which is then broken down to release iron as it is needed, in a process known as ferritinophagy.

Cells use several systems to break down proteins, one of which, called autophagy, has been linked to ferritinophagy. During autophagy, a bubble-like structure called an autophagosome engulfs proteins that need to be removed and delivers them to a compartment in the cell where they can be broken down. In 2014, researchers showed that a protein called NCOA4 on the surface of autophagosomes targets ferritin for destruction. When iron levels are high in the cell, the amount of NCOA4 on the autophagosomes is low. This leads to fewer ferritin molecules being broken down. In contrast, low iron levels lead to an increase of NCOA4 on autophagosomes, which promotes ferritinophagy and increases the amount of iron in the cell.

Now, Mancias, Vaites et al—including several of the researchers involved in the 2014 work—investigate the role of NCOA4 in ferritinophagy in more detail. Biochemical experiments revealed that a region of NCOA4 directly interacts with a particular subunit of ferritin and this interaction is necessary to deliver ferritin to autophagosomes.

Mancias, Vaites et al. then used laboratory grown-cells to investigate why the amount of NCOA4 changes in response to the amount of iron in the cell. The experiments show the amount of NCOA4 varies depending on whether it interacts with another protein called HERC2, which targets proteins for destruction by a structure called the proteasome. HERC2 only binds to NCOA4 when iron levels are high, which leads to NCOA4 being broken down by the proteasome. When iron levels are low, HERC2 does not interact with NCOA4. The presence of more NCOA4 then leads to more ferritinophagy, and so increases the amount of iron in the cell.

Mancias, Vaites et al. also found that red blood cells, which depend highly on iron, do not develop properly in zebrafish that have lower amounts of the NCOA4 protein. Further work is needed to see whether NCOA4 is also important for the development of other cells and tissues.

process (*Radisky and Kaplan, 1998*; *Asano et al., 2011*). Through quantitative proteomic analysis of purified autophagosomes, we recently identified NCOA4 as the selective autophagy receptor that targets ferritin to the autophagosome, providing the first mechanistic explanation of how ferritinophagy occurs (*Mancias et al., 2014*). We further showed that NCOA4-mediated ferritinophagy is important for maintaining cellular iron homeostasis. NCOA4 was subsequently identified as an ubiquitylation target when the VPS34 autophagy activator is inhibited and also was found to promote ferritinophagy (*Dowdle et al., 2014*).

Three central questions emerge from this work. First, what molecular determinants drive the pathway and are the interactions between NCOA4 and ferritin direct? Interestingly, NCOA4 has been reported to bind FTH1, but not FTL, in cell extracts (*Dowdle et al., 2014*); however, the biochemical basis of this specificity is unclear, especially given that FTH1 and FTL are 53% identical and 66% similar in primary sequence. In most cell types, ferritin complexes are composed of both heavy and light chains. The identification of point mutants that specifically abolish these interactions could facilitate a further dissection of NCOA4 function.

Second, given that NCOA4 levels appear to control flux through the ferritinophagy pathway, what mechanisms control the abundance of NCOA4? We previously reported that NCOA4 levels are altered by intracellular iron status (*Mancias et al., 2014*); when iron levels are high, NCOA4 abundance is low, thereby promoting ferritin accumulation and iron capture. But when iron is low, NCOA4 levels increase to promote ferritinophagy. However, our understanding of the mechanisms underlying these processes is limited and is complicated by the fact that the pool of NCOA4 participating in targeting ferritin to autophagosomes is apparently degraded within lysosomes. Thus, understanding the mechanisms that control NCOA4 abundance could help unravel the physiological underpinnings of flux through the ferritinophagy pathway, with implications for understanding iron utilization. Our recent work revealed

HERC2, a large multi-domain homologous to E6AP carboxy terminus (HECT) E3 ubiquitin ligase, as an NCOA4-interacting protein (*Mancias et al., 2014*). HERC2 is reported to associate with numerous proteins and likely has multiple ubiquitylation targets, including E6AP and proteins linked to DNA damage (*Bekker-Jensen et al., 2010*; *Martinez-Noel et al., 2012*; *Moroishi et al., 2014*; *Galligan et al., 2015*). Intriguingly, HERC2 associates with the SCF$^{FBLX5}$ ubiquitin ligase (*Tan et al., 2013*) and has recently been implicated in the basal turnover of FBXL5 (*Moroishi et al., 2014*), a key iron-sensing protein that promotes turnover of the ferritin translational repressor IRP2 (Iron Regulatory Protein 2, also known as IREB2) when iron is high (*Salahudeen et al., 2009*; *Vashisht et al., 2009*). However, the precise role that HERC2, iron, and the ubiquitin system play in regulating NCOA4, and how this is related to NCOA4 autophagic turnover during ferritinophagy remains unexplored.

Third, what is the physiological role of NCOA4 in cellular processes that are highly dependent upon iron availability? Initial links between NCOA4 and processes with a requirement for high iron availability come from expression studies where *ncoa4* mRNA is high at sites of erythropoiesis during zebrafish development (*Weber et al., 2005*). Moreover, recent transcriptomic analysis shows significant upregulation of *NCOA4* at the orthochromatic erythroblast stage of erythroid differentiation in humans, the stage associated with massive heme and hemoglobin synthesis, a highly iron dependent process (*An et al., 2014*). Prior studies have demonstrated defects in erythroid differentiation upon deletion of canonical autophagy genes (*Cao et al., 2015*); however, these studies focused on an inability to clear mitochondria during erythroid maturation due to dysfunctional mitophagy (*Mortensen et al., 2010*; *Li-Harms et al., 2015*). While it is clear that iron obtained via holo-transferrin endocytosis is used for heme synthesis during erythroid differentiation, there is debate as to the intracellular itinerary of this iron once liberated from transferrin (*Lane et al., 2015*). There is evidence that transferrin-liberated endosomal iron is transferred directly to mitochondria (*Sheftel et al., 2007*); however, additional studies show that ferritin-sequestered iron is also utilized for heme synthesis (*Vaisman et al., 1997*).

Here, we use in-depth biochemical and cell culture studies, as well as the zebrafish system to dissect the roles of NCOA4, Ferritin, and HERC2 in mediating ferritinophagy and the role of ferritinophagy in erythropoiesis. We show that NCOA4 interacts directly with FTH1 via a conserved NCOA4 C-terminal domain and a key conserved residue on FTH1. Mutation at these binding sites abrogates binding in vivo and abolishes ferritinophagy. We show that the HERC2 ligase uses its 'CUL7-homology domain' to recognize NCOA4 under high iron conditions to mediate NCOA4 turnover via the ubiquitin-proteasome system, thereby reducing the steady-state NCOA4 levels and increasing ferritin for iron capture. Surprisingly, we find that this same C-terminal domain within NCOA4 binds iron and the iron-bound state of NCOA4 determines HERC2 binding, suggesting an iron-dependent switch in NCOA4 turnover. Finally, we show that NCOA4 is important for erythropoiesis in vivo given its role in mobilizing iron from ferritin for use in heme synthesis. This study establishes the importance of NCOA4 as a critical regulator of cellular and organismal iron metabolism and reveals the mechanistic underpinnings of its iron-dependent regulation.

## Results

### NCOA4 interacts with ferritin via a conserved C-terminal domain

There is little structural information available for NCOA4 apart from predicted coiled coil domains at the N-terminus. While NCOA4 orthologs exist throughout metazoans, there is minimal sequence homology within the proteome. Sequence alignment, secondary structure prediction, and tertiary structure prediction were used to design NCOA4 fragments for identification of the ferritin-binding domain (*Figure 1A*). The N-terminus of NCOA4 consists of predicted coiled coil domains that have been previously shown to mediate oligomerization of NCOA4 (*Monaco et al., 2001*). This domain is present in both NCOA4 splice variants, encoding a 614-residue α isoform and a 287-residue β isoform (*Alen et al., 1999*). We therefore tested binding of purified apoferritin from horse spleen (containing both FTH1 and FTL) in vitro to recombinant full-length Myc-tagged NCOA4α, NCOA4β (to rule out a folded motif consisting of the N-terminus and a short portion of the C-terminus), NCOA4-N-terminus (NCOA4$^{1–245}$), and NCOA4-C-terminus (NCOA4$^{235–614}$). Ferritin associated with both NCOA4α and the NCOA4 C-terminal fragment but not with NCOA4β or the NCOA4 N-terminal fragment (*Figure 1B*). Further truncation constructs were designed based on secondary structure prediction and ferritin binding in vitro was mapped to NCOA4 amino acids 383–522 (NCOA4$^{383–522}$, *Figure 1A,C,D*).

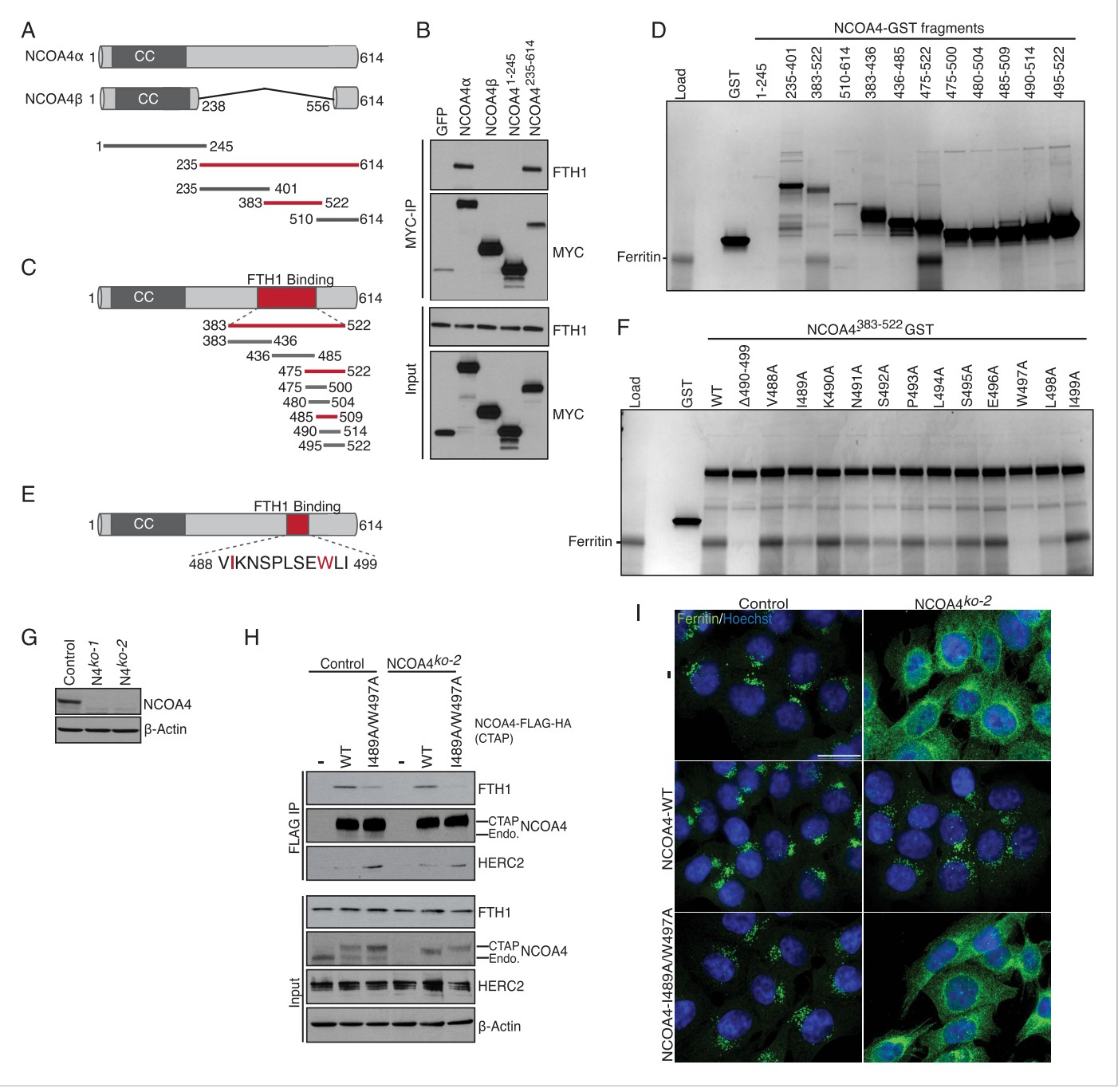

**Figure 1**. NCOA4 recognizes FTH1 via a conserved C-terminal domain. (**A**) Schematic of NCOA4 truncation constructs. (**B**) Horse spleen apoferritin was mixed with in vitro translated MYC-NCOA4 fragments, immunoprecipitated with anti-MYC (MYC-IP) and immunoblotted with the indicated antibodies. (**C**) Schematic of NCOA4$^{383–522}$ truncation constructs. (**D**) GST pulldown assay of recombinant NCOA4-GST fragments as indicated mixed with ferritin. Bound ferritin was analyzed by 4–20% SDS-PAGE and Coomassie blue staining. Load is 5% of ferritin input. (**E**) Schematic and sequence of NCOA4 point mutations designed within NCOA4$^{383–522}$. (**F**) GST pulldown assay of recombinant NCOA4-GST fragments as indicated mixed with ferritin. Bound ferritin was analyzed by 4–20% SDS-PAGE and Coomassie blue staining. Load is 5% of ferritin input. (**G**) CRISPR/Cas9-mediated depletion of NCOA4 expression in HCT116 cells. Two independent NCOA4 knockout clones were generated following targeting of NCOA4 exon 2. A CRISPR non-targeting control cell line was also established. (**H**) Control or NCOA4$^{ko-2}$ cells were transduced with NCOA4$^{WT}$ or NCOA4$^{I489A/W497A}$ lentivirus and stable cell lines were generated. Endogenous FTH1 binding was evaluated by immunoblot following FLAG affinity purification of WT or mutant NCOA4. (**I**) NCOA4$^{I489A/W497A}$ attenuates FTH1 localization in lysosomes following iron chelation. HCT116 control or NCOA4 knockout cells were plated on glass coverslips and treated with FAC for 14 hr. To promote ferritin accumulation in lysosomes, cells were then washed and treated with DFO plus lysosomal protease inhibitors E64-d and Pepstatin A for 6 hr. Cells were fixed, stained with ferritin antibody and visualized by confocal microscopy. Scale bar, 20 μm.

*Figure 1. continued on next page*

*Figure 1. Continued*

The following figure supplement is available for figure 1:

**Figure supplement 1**. NCOA4 interacts with ferritin via a C-terminal domain and promotes lysosomal ferritin accumulation upon iron depletion.

This portion of NCOA4 is predicted to contain four α-helices and constitutes a discrete sub-domain of NCOA4 not present in NCOA4β. To determine if a discrete portion of this region binds ferritin, we made further truncation constructs and demonstrated binding to amino acids 475–522 (*Figure 1C,D*). There was a significant loss of binding efficiency to ferritin when further truncation constructs were tested; however, residual binding was seen with amino acids 485–509 (*Figure 1D*, *Figure 1—figure supplement 1A*). Given the overlapping nature of truncation constructs in this experiment, we tested whether deletion of amino acids 490–499 within the NCOA4$^{383–522}$ construct affected binding in vitro. This construct completely abrogated binding (*Figure 1F*). Alanine scanning mutagenesis across this region identified W497 and to a lesser extent I489, S492, L494, and L498 as important residues for ferritin binding (*Figure 1E,F*). Mutation of I489 and W497 (NCOA4$^{I489/W497A}$) in the context of NCOA4$^{383–522}$ abrogated binding to ferritin (*Figure 1—figure supplement 1B*). Thus, NCOA4 associates directly with ferritin in vitro and employs sequences in a predicted helical domain for this interaction.

## Binding of NCOA4 to endogenous FTH1 requires I489/W497 and is required for ferritinophagy

We next examined the ability of NCOA4$^{I489A/W497A}$ to bind endogenous ferritin in HCT116 cells and evaluated whether mutant NCOA4 was capable of facilitating ferritin delivery to lysosomes upon iron depletion. Because dimerization with endogenous NCOA4 could confound analysis of mutants, we utilized CRISPR/Cas9-mediated genome editing (*Ran et al., 2013*) to engineer NCOA4-deficient clonal HCT116 cell lines. Two independent clones were obtained and analyzed for NCOA4 protein expression (*Figure 1G*). To determine whether mutant NCOA4 binds ferritin in cells, we stably expressed C-terminally tagged (CTAP) wild type or NCOA4$^{I489A/W497A}$ in control or NCOA4-deficient HCT116 cells, followed by FLAG affinity purification to purify tagged NCOA4. Consistent with in vitro data, ferritin binding to NCOA4$^{I489A/W497A}$ (as monitored by FTH1 immunoblotting) is abrogated in NCOA4-deficient cells, and a significant reduction in binding to ferritin is observed in control cells expressing endogenous NCOA4, apparently reflecting oligomerization of ectopically expressed NCOA4$^{I489A/W497A}$ with endogenous NCOA4 present in the immune complex (*Figure 1H*).

Given that NCOA4$^{I489A/W497A}$ displayed a dramatic decrease in ferritin binding when expressed in cells, we next evaluated whether cells expressing this mutant undergo ferritinophagy upon iron depletion. Control or NCOA4-deficient cells stably expressing wild type or NCOA4$^{I489A/W497A}$ were loaded with ferric ammonium citrate (FAC) for 14 hr, followed by iron chelation using deferoxamine (DFO) for 6 hr in the presence of lysosomal protease inhibitors. While NCOA4-deficient cells expressing wild type NCOA4 exhibit punctate lysosomal ferritin localization as shown previously (*Dowdle et al., 2014*; *Mancias et al., 2014*) cells expressing NCOA4$^{I489A/W497A}$ exhibit diffuse cytoplasmic ferritin staining, indicating defective ferritinophagy in the presence of mutant NCOA4 (*Figure 1I*, *Figure 1—figure supplement 1C*). Our data indicates that NCOA4$^{383–522}$, including hydrophobic residues I489 and W497, is required for association with ferritin and productive ferritinophagy in cells.

## FTH1 R23 is essential for ferritin association with NCOA4 in vitro and in cells

Ferritin complexes in vivo are composed of 24 subunits of FTH1 and FTL, with varying ratios depending on the cell type. We initially identified both FTH1 and FTL as binding partners of NCOA4 and confirmed a reciprocal interaction by FTH1 pulldown (*Mancias et al., 2014*). While Dowdle et al. recently showed that NCOA4 appears to selectively recognize the FTH1 subunit of ferritin in vivo (*Dowdle et al., 2014*), it remains unclear whether the NCOA4-ferritin interaction occurs in a direct manner or requires intermediate factors for binding. While the above experiments with purified ferritin from horse spleen and recombinant NCOA4 suggested a direct interaction, we subsequently produced

recombinant FTH1-only and FTL-only ferritin complexes and tested for direct interactions with recombinant NCOA4. NCOA4 specifically recognizes recombinant FTH1, confirming the prior specificity and also demonstrating that NCOA4 recognizes FTH1 directly (*Figure 2A*). As NCOA4 interacts with the 450 kDa ferritin complex (*Dowdle et al., 2014*), we predicted that NCOA4 binds to the surface of ferritin and that differences in FTH1 vs FTL surface residues dictate NCOA4 specificity for FTH1. We identified 16 FTH1 surface residues that were conserved among FTH1 orthologs but not FTL (*Figure 2B*) (*Lawson et al., 1991*; *Hempstead et al., 1997*). We performed alanine mutagenesis of these 16 residues individually, expressed them recombinantly in *Escherichia coli*, purified 450 kDa complexes for all mutants as determined by gel filtration analysis, and tested binding to NCOA4 in NCOA4-GST pulldowns. Critically, FTH1$^{R23A}$ mutation abrogated binding to NCOA4$^{383-522}$ but none of the other residues tested had a negative effect on NCOA4-FTH1 binding in these assays (*Figure 2C*). Importantly, purified FTH1$^{R23A}$ forms 450 kDa ferritin complexes, similar to FTH1$^{WT}$ (*Figure 2—figure supplement 1A,B*).

To determine whether FTH1 R23 is required to support NCOA4 association with ferritin complexes in cells, we first generated FTH1-deficient cells using CRISPR/Cas9-mediated genome editing. Three independent FTH1-deficient HCT116 cell lines were obtained, as evidenced by loss of FTH1 protein expression and PCR genotyping (*Figure 2D*, *Figure 2—figure supplement 1C*). We stably expressed N-terminally FLAG-HA tagged (NTAP) wild type and FTH1$^{R23A}$ in control or FTH1-deficient HCT116 cells, followed by FLAG affinity purification to isolate FLAG-FTH1 complexes and examine endogenous NCOA4 binding. While FTH1$^{R23A}$ is unable to associate with NCOA4, binding to FTL is maintained, indicating that mutant FTH1 is capable of forming FTH/FTL complexes (*Figure 2E*, *Figure 2—figure supplement 1C*). Moreover, GFP-FTH1$^{R23A}$ expressed in FTH1-deficient cells fails to localize to punctate lysosomal structures, consistent with an inability to bind the NCOA4 cargo receptor (*Figure 2F*). Thus, binding of NCOA4 to FTH1 involves a basic residue located on the surface of the complex but not present in FTL.

## Iron-dependent NCOA4 downregulation by the HERC2 E3 ubiquitin ligase

Our previous work revealed that NCOA4 delivers ferritin complexes to the autophagosome for degradation (*Mancias et al., 2014*). Since aberrant degradation of ferritin in lysosomes would be deleterious to the cell when intracellular iron is high, we next sought to elucidate the mechanism whereby cells suppress NCOA4-mediated ferritinophagy under iron-replete conditions. We previously noted that NCOA4 levels are altered by iron status; when iron is high, NCOA4 abundance is low thereby promoting ferritin accumulation, and when iron is low, NCOA4 levels are high, thereby promoting ferritinophagy (*Mancias et al., 2014*). We noted that previous interaction proteomics revealed the large HECT E3 ligase HERC2 as a high confidence NCOA4 interacting protein (*Mancias et al., 2014*). We hypothesized that HERC2 may regulate NCOA4 levels via the ubiquitin-proteasome system. Indeed, previous proteomics experiments identified multiple ubiquitylation sites on NCOA4 (*Kim et al., 2011*; *Dowdle et al., 2014*). We further reasoned that the iron-dependent regulation of NCOA4 levels may be a result of iron-dependent recognition of NCOA4 by HERC2. Therefore we first examined the binding of endogenous HERC2 to NCOA4 in cells under iron replete and deficient conditions. Cells were treated with iron chelator (DFO), or loaded with excess iron (FAC) followed by HERC2 or NCOA4 immunoprecipitation (IP). Immunoblot analysis of endogenous NCOA4 revealed an interaction with HERC2 under basal cell culture conditions where iron is replete and that this interaction was enhanced under excess iron conditions. Conversely, HERC2 binding to NCOA4 was severely attenuated following iron chelation (*Figure 3A*). We also examined endogenous HERC2 binding to stably expressed NCOA4-FLAG-HA (CTAP) tagged protein under iron replete and deficient conditions. Immunoblot analysis of endogenous HERC2 revealed an interaction in iron replete and excess iron conditions, while HERC2 binding to NCOA4 was attenuated following iron chelation (*Figure 3—figure supplement 1A*).

To define the region of HERC2 that recognizes NCOA4, we sub-cloned and expressed six fragments spanning the HERC2 open reading frame as depicted in *Figure 3B* (*Bekker-Jensen et al., 2010*). Endogenous NCOA4 associated with Fragment 3 (HERC2$^{F3}$, amino acids 1700–2800; *Figure 3C,D*), comprised of a homology region shared by HERC2 and Mindbomb E3 ligases and the CUL7 CPH domain (conserved domain in CUL7, PARC, and HERC2 proteins) (*Itoh et al., 2003*; *Kasper et al., 2006*; *Kaustov et al., 2007*). Further mapping of the HERC2 sequence between amino acids 1700–2800 revealed a strong interaction between endogenous NCOA4 and HERC2$^{F3-6}$,

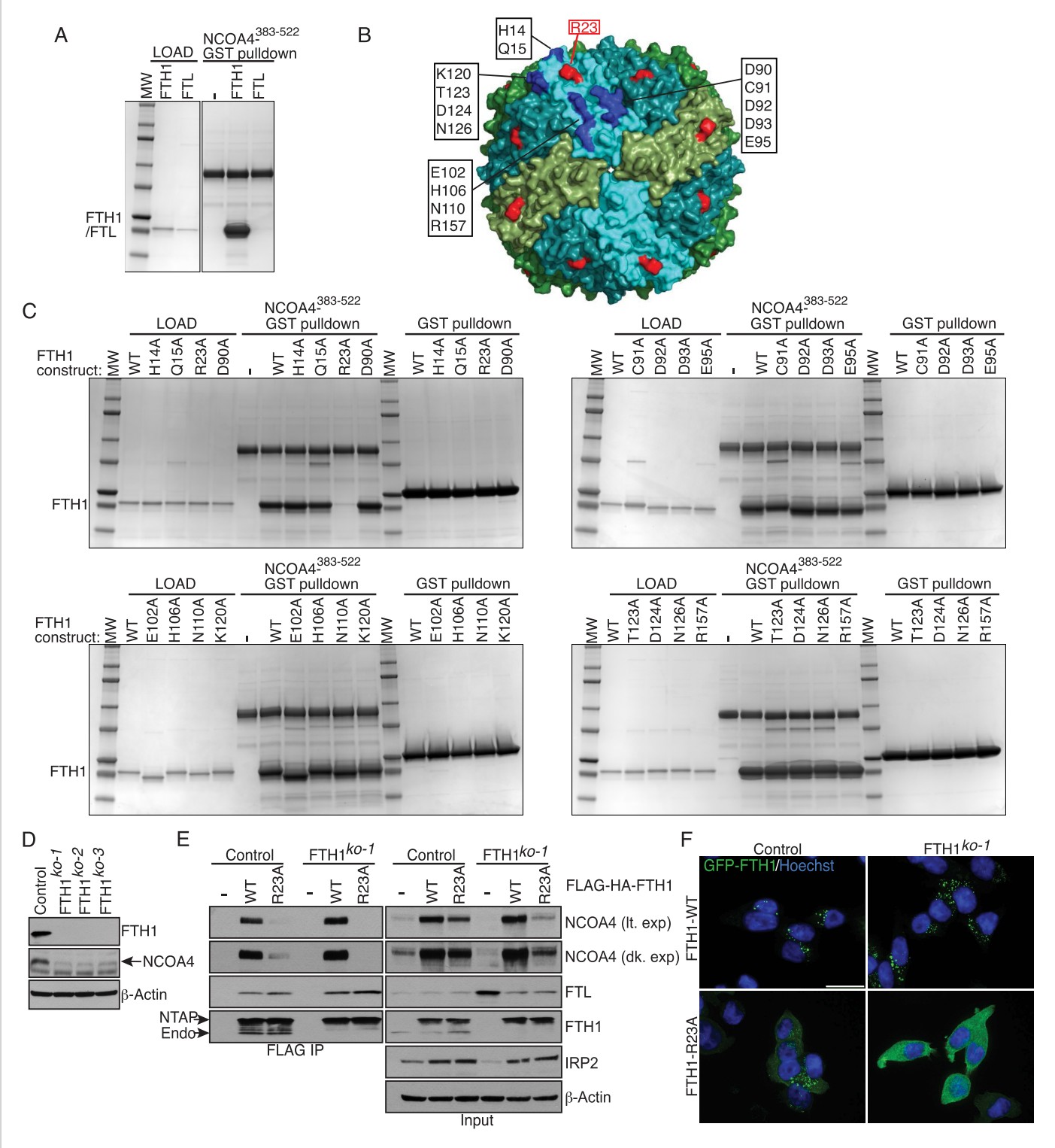

**Figure 2**. FTH1$^{R23}$ is essential for NCOA4 binding and ferritinophagy. (**A**) GST pulldown assay of recombinant NCOA4$^{383–522}$-GST mixed with recombinant FTH1-only or FTL-only complexes. Bound FTH1 or FTL was analyzed by 4–20% SDS-PAGE and Coomassie blue staining. Load is 5% of FTH1 or FTL input. (**B**) Space-filling model of FTH1 ferritin complex (PDBid: 3AJO) with individual subunits colored in greens and blues, 16 conserved surface residues are highlighted in dark blue or red (R23) and labeled as indicated. (**C**) GST pulldown assay of recombinant NCOA4$^{383–522}$-GST or GST alone mixed with recombinant wild type (WT) and point mutant (as indicated) FTH1-only complexes. Bound FTH1 or FTL was analyzed by 4–20% SDS-PAGE and Coomassie blue staining. Load is 5% of FTH1 input. (**D**) CRISPR/Cas9-mediated depletion of FTH1 expression in HCT116 cells. Three independent FTH1 knockout clones were generated following targeting of FTH1 exon 1. (**E**) Control or FTH1$^{ko−1}$ cells were transduced with FTH1$^{WT}$ or FTH1$^{R23A}$ lentivirus and stable cell

*Figure 2. continued on next page*

Figure 2. Continued
lines were generated. Endogenous NCOA4 and FTL binding was evaluated by immunoblot following FLAG affinity purification of WT or mutant FTH1.
(F) Mutation of FTH1[R23] attenuates GFP-FTH1 localization in lysosomes. HCT116 control or FTH1 knockout cells were plated on glass coverslips and
treated with Bafilomycin for 6 hr to prevent lysosomal degradation of ferritin. Cells were fixed and visualized for GFP-FTH1 localization by confocal
microscopy. Scale bar, 20 µm.
The following figure supplement is available for figure 2:

Figure supplement 1. FTH1 R23A forms functional ferritin cages and abrogates NCOA4 binding.

corresponding to the CUL7 homology domain and an unstructured region of HERC2 (residues 2540–2700, *Figure 3E*). Iron chelation also abrogated binding of NCOA4 to exogenously expressed HERC2 (*Figure 3F*).

The finding that HERC2 predominantly associates with NCOA4 when intracellular iron is high led us to examine whether HERC2 modulates NCOA4 stability, thereby reducing flux through the ferritinophagy pathway. Our previous work revealed a reduction in NCOA4 protein expression upon excess iron loading (*Mancias et al., 2014*). Knockdown of HERC2 using two independent siRNAs resulted in accumulation of NCOA4 protein (*Figure 3G*). Furthermore, siRNA-mediated HERC2 depletion promoted a significant increase in NCOA4 half-life (*Figure 3H,I*). Further supporting a role for HERC2 in regulating NCOA4 stability in the presence of high intracellular iron, overexpression of HERC2[F3–6] abrogated NCOA4 loss following excess iron loading likely through interfering with the binding of NCOA4 to endogenous HERC2 (*Figure 3J*). Similarly, HERC2 knockdown stabilized NCOA4 following FAC treatment (*Figure 3—figure supplement 1B*). Notably, HERC2 depletion did not result in complete stabilization of NCOA4 turnover. Consistently, pre-treatment of cells with the proteasome inhibitor Bortezomib (Btz) only partly rescued NCOA4 turnover in response to cycloheximide (CHX) treatment (*Figure 3—figure supplement 1C*). Likewise, pre-treatment of cells with the autophagy inhibitor bafilomycin (Baf) only partly rescued NCOA4 turnover in response to CHX treatment (*Figure 3—figure supplement 1D*). We determined that in addition to HERC2-dependent turnover, NCOA4 also undergoes turnover via basal autophagy. Under iron deficient conditions, despite continued autophagic turnover, the apparent lack of HERC2 action on NCOA4 leads to a net increase in NCOA4 protein level thereby promoting ferritinophagy to release iron from ferritin for use by the cell.

## NCOA4 binds iron and HERC2 association is dependent on NCOA4 iron occupancy

Using recombinant proteins, we subsequently mapped HERC2 binding to NCOA4[383–522], the same region as is necessary for FTH1 binding (*Figure 4A*). Efforts to map the interaction to a sub-domain of NCOA4[383–522] were unsuccessful and suggest that the binding requires a folded epitope on NCOA4. Notably, mutation of residues I489 and W497 had no effect on binding to HERC2 (*Figure 1H*) suggesting a HERC2 binding site distinct from that for FTH1. We further mapped the binding on HERC2 to the minimal CUL7 domain (amino acids 2553–2639; *Figure 4B*). Similar to in vivo studies, this interaction was dependent on a chelatable iron ion as DFO attenuated binding in vitro (*Figure 4C*). The dependence of the HERC2-NCOA4 interaction on iron levels suggests that either the complex or the individual components bind an iron molecule. While neither NCOA4 nor the HERC2 CUL7 domain have predicted iron binding sites (*Passerini et al., 2011*), inductively coupled plasma mass spectrometry (ICP-MS) analysis showed that iron co-purified with a recombinant fragment of NCOA4[383–509], but not the HERC2 CUL7 domain or buffer alone (*Figure 4C*, *Figure 4—figure supplement 1A,B*). Thus, NCOA4 (through residues 383–509) can function as an iron-binding protein and its ability to interact with HERC2 in cells is anticipated to depend on the level of bioavailable iron.

## NCOA4 regulates erythroid differentiation in cell culture and zebrafish models

Previous studies suggest that NCOA4 has a role in erythropoiesis due to the timing and pattern of expression (*Weber et al., 2005*; *Nilsson et al., 2009*; *An et al., 2014*). We therefore examined the role

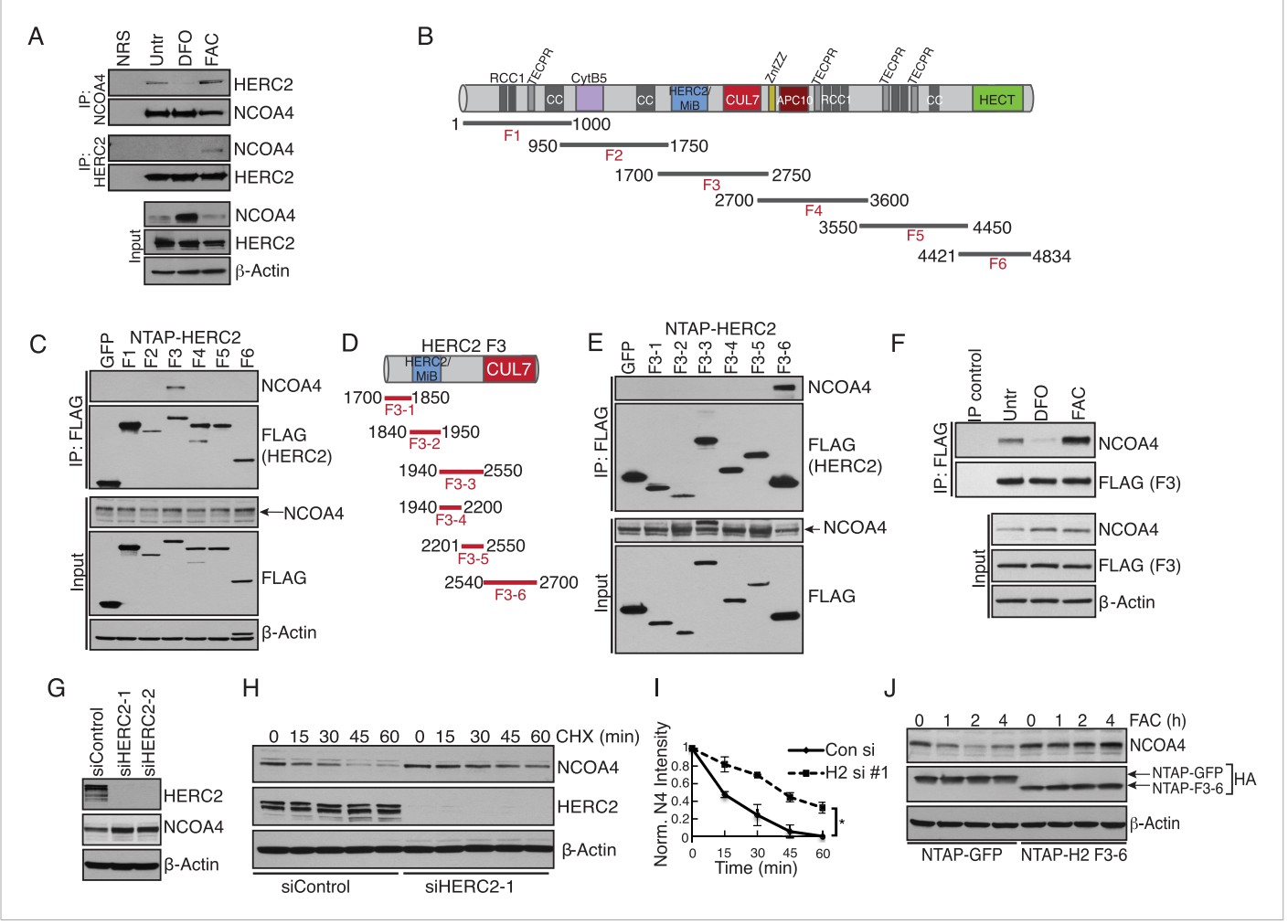

**Figure 3**. Iron-dependent NCOA4 turnover via the HERC2 E3 Ubiquitin Ligase. (**A**) Endogenous IP of NCOA4 from 293T cells treated with DFO or FAC for 6 hr. Lysate input for control IgG IP is the untreated sample. HERC2 and NCOA4 binding was evaluated by immunoblot. (**B**) Schematic of HERC2 functional domains. To interrogate NCOA4 binding to HERC2, six fragments spanning the HERC2 protein (F1-6) were subcloned and expressed in 293T cells. (**C**) Endogenous NCOA4 binding to NTAP-HERC2 fragments. Extracts from 293T cells stably expressing the indicated proteins were immunoprecipitated with anti-Flag (Flag-IP) and immunoblotted with the indicated antibodies. (**D**) Identification of the minimal HERC2 domain responsible for NCOA4 binding. HERC2$^{F3}$ was divided into six sub-fragments as indicated. (**E**) NCOA4 binds the CUL7 (CPH) homology domain of HERC2. HERC2 FLAG-tagged fragments 3-1 through 3–6 were expressed in 293T cells followed by Flag-IP, and immunoblot with the indicated antibodies. (**F**) HERC2 CUL7 domain binds endogenous NCOA4 in an iron-dependent manner. FLAG-tagged HERC2$^{F3}$ was expressed in 293T cells, followed by DFO or FAC treatment for 9 hr and Flag-IP as in (**C**). Lysate input for the control IP (normal mouse serum) corresponds to the untreated sample. (**G**) siRNA-mediated knockdown of HERC2 expression promotes NCOA4 protein accumulation. U2OS cells were transfected with control or HERC2-specific siRNAs for 72 hr, harvested, and subjected to SDS-PAGE and immunoblot with the indicated antibodies to evaluate HERC2 knockdown and NCOA4 abundance. (**H**) NCOA4 half-life is extended upon depletion of HERC2. Immunoblot of NCOA4 protein levels in U2OS cells following control or HERC2 siRNA delivery and cycloheximide treatment (CHX) as indicated. (**I**) Quantification of NCOA4 protein levels in 2 independent biological CHX experiments. Error bars represent +/− standard deviation, *, p < 0.05 by two-tailed unpaired t-test at each time point. (**J**) Expression of HERC2$^{2540–2700}$ abrogates iron-mediated NCOA4 downregulation. 293T cells were transfected with FLAG-tagged HERC2$^{F3–6}$ or FLAG-tagged GFP as a control, followed by FAC treatment as indicated. NCOA4 protein level was evaluated by immunoblot.

The following figure supplement is available for figure 3:

**Figure supplement 1**. HERC2 regulates NCOA4 turnover.

of NCOA4 in a K562 erythroleukemia cell culture model of hemin-induced erythroid differentiation. NCOA4 shRNA-mediated knockdown abrogated increases in hemoglobin mRNA and protein levels in comparison to shGFP lines upon hemin differentiation (***Figure 5A–C***). Hemoglobinization was likewise

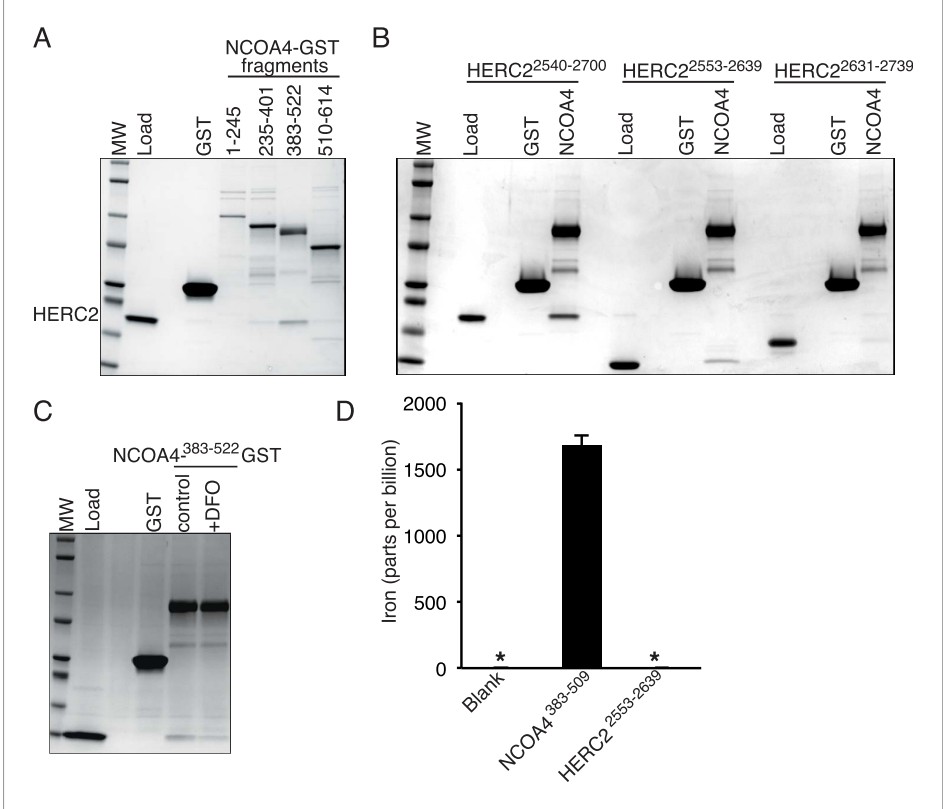

**Figure 4**. NCOA4 is an iron-binding protein. (**A**) GST pulldown assay of recombinant NCOA4-GST fragments as indicated mixed with recombinant HERC2$^{2540-2700}$. Bound HERC2 was analyzed by 4–20% SDS-PAGE and Coomassie blue staining. Load is 5% of HERC2 input. (**B**) GST pulldown assay of recombinant NCOA4$^{383-522}$-GST mixed with recombinant HERC2 fragments as indicated. Bound HERC2 was analyzed by 4–20% SDS-PAGE and Coomassie blue staining. Load is 5% of HERC2 input. (**C**) GST pulldown assay of recombinant NCOA4$^{383-522}$-GST pre-treated as indicated with DFO and mixed with recombinant HERC2 CUL7 domain (amino acids 2553–2639). Bound HERC2 was analyzed by 4–20% SDS-PAGE and Coomassie blue staining. Load is 5% of HERC2 input. (**D**) NCOA4 NCOA4$^{383-522}$-GST or the HERC2$^{2553-2639}$ (CUL7 domain) was expressed in *Escherichia coli* and the amount of co-purifying iron was measured by means of Inductively coupled plasma mass spectrometry (ICP-MS).

The following figure supplement is available for figure 4:

**Figure supplement 1**. Evaluating NCOA4 iron binding.

severely affected. The dark brown coloration of NCOA4 shRNA pellets suggests accumulation of iron-laden ferritin with inability to release iron for subsequent heme synthesis (*Figure 5D*).

We next confirmed *ncoa4* is expressed at sites of hematopoiesis in zebrafish as previously published (*Weber et al., 2005*) including in the intermediate cell mass (ICM) at 24 hpf (hours post-fertilization), a structure that is equivalent to the mouse yolk sac E7.5–10.0. At 48 and 72 hpf, *ncoa4* is expressed in circulating erythrocytes visible in the yolk sac, heart, and liver (L) (*Figure 5E*). Consistent with its expression in sites of hematopoiesis and circulating erythrocytes, morpholino knockdown of *ncoa4* severely disrupts erythropoiesis as visualized at 30 hpf in *globin-LCR:eGFP* transgenic zebrafish, which express GFP in erythroid cells (*Ganis et al., 2012*) (*Figure 5F*). Morphants exhibit severe reduction in circulating GFP-labeled erythroid cells (*Figure 5G*). Furthermore, hemoglobinization as visualized by *o*-dianisidine staining was severely reduced using two independent morpholinos to *ncoa4* (*Figure 5H*). Taken together these findings suggest that during erythroid differentiation, the majority of iron obtained via endocytosed holo-transferrin is stored in ferritin before mobilization via NCOA4-mediated ferritinophagy for heme synthesis.

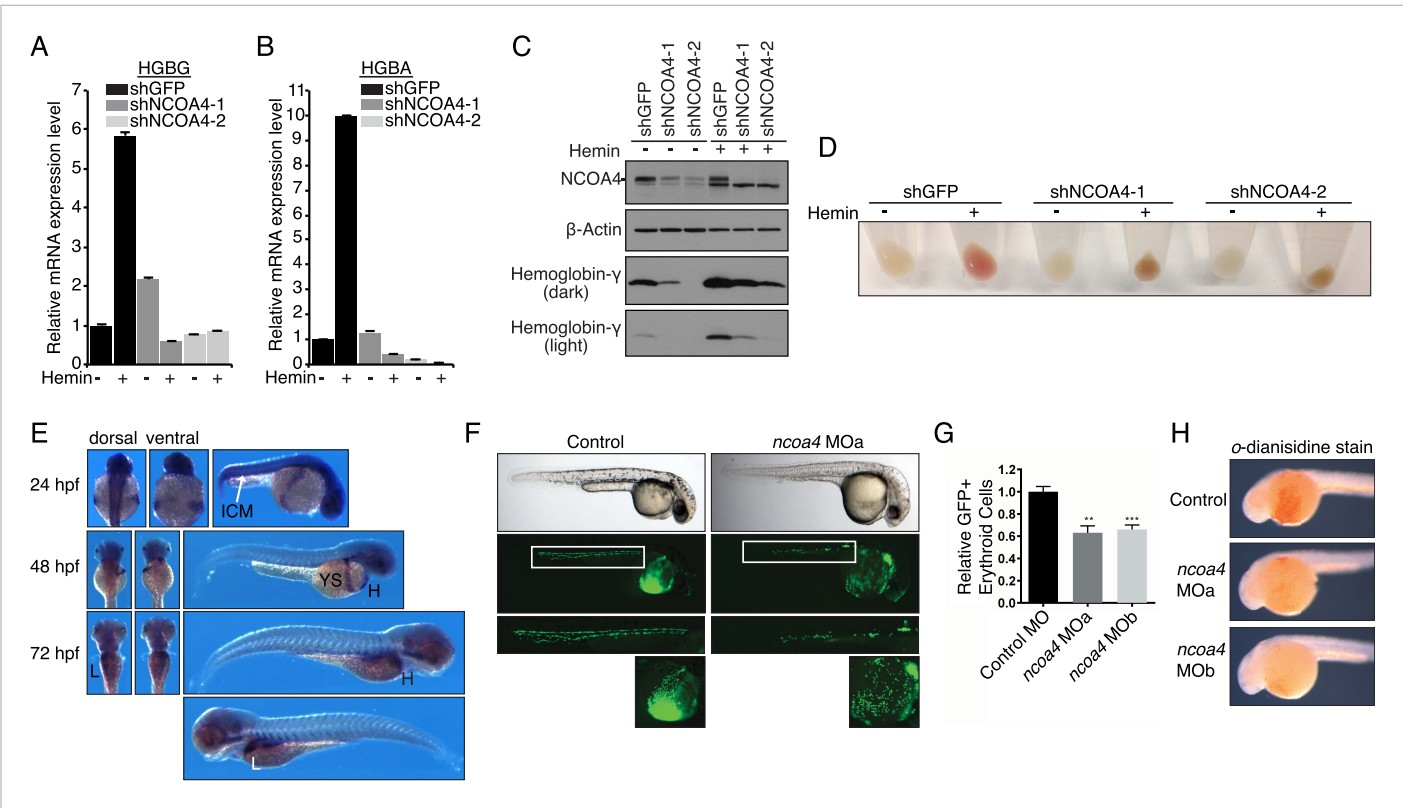

**Figure 5**. Knockdown of *ncoa4* impacts erythropoiesis in zebrafish and tissue culture cells. (**A**, **B**) Expression of *HGBG* (**A**), *HGBA* (**B**) determined by quantitative RT-PCR in K562 cells expressing a control (shGFP) or two independent shRNAs targeting *NCOA4*. Hemin treatment (25 µM, 72 hr) as indicated. (**C**) Immunoblot analysis of K562 cells as in (**A**) and (**B**) with indicated antibodies. Hemin treatment (25 µM, 72 hr) as indicated. (**D**) Appearance of K562 pellets as in (**A**, **B**, and **C**). Red coloration indicates appropriate hemoglobinization of cells after hemin differentiation. Brown coloration indicates accumulation of iron without hemoglobinization. (**E**) Expression of *ncoa4* in circulating erythroid cells relative to sites of primitive hematopoiesis. Abbreviations: hpf = hours post-fertilization, ICM = intermediate cell mass, YS = yolk sac, H = heart, L = liver. (**F**) Circulating RBCs are visualized in *globin-LCR:eGFP* erythrocyte reporter zebrafish at 30 hpf. Morpholino-mediated knockdown of *ncoa4* severely disrupts erythropoiesis (n > 30 each condition). Inset shows erythrocytes circulating in the caudal artery and caudal vein plexus. (**G**) FACS quantification of erythrocytes in *globin-LCR:eGFP* reporter zebrafish at 30 hpf following control or *ncoa4*-morpholino-mediated knockdown (*ncoa4* MOa and MOb). ***p < 0.001 and **p < 0.002 by two-tailed unpaired t-test. (**H**) *o*-dianisidine staining (brown) for hemoglobinized red cells in zebrafish embryos. Embryos were grown from zygotes injected at the one- to two-cell stage with *ncoa4*-targeting morpholinos (MOa and MOb) or control injected zygotes (control). *Ncoa4* MOa: 15/16 embryos with diminished staining in comparison to control. *Ncoa4* MOb: 18/22 embryos with diminished staining in comparison to control.

## Discussion

Our data support a model whereby direct binding of NCOA4 to FTH1 is critical for ferritinophagy. This interaction appears to be present basally in all cells examined suggesting that some basal level of ferritinophagy is necessary to maintain iron homeostasis under standard tissue culture conditions. As discussed below, higher levels of NCOA4 during periods of iron depletion increase ferritinophagy, suggesting that flux through the ferritinophagy pathway is controlled by NCOA4 abundance. While we have not directly ascertained the stoichiometry of the NCOA4-FTH1 complex, our data utilizing the FTH1[R23A] mutant in control cells suggests it is far less than a 1:1 stoichiometry. Lentiviral-based expression of FTH1[R23A] in HCT116 cells leads to expression of FTH1[R23A] above endogenous FTH1 levels, with the predominance of ferritin complexes being composed of primarily the FTH1[R23A] subunit and sub-stoichiometry levels of endogenous wild type FTH1 that remains competent to bind NCOA4. Despite this, GFP-FTH1[R23A] labeled ferritin complexes still localized to punctate lysosomes suggesting that one or a few wild type FTH1 molecules in the complex is sufficient for targeting. This would be consistent with a model whereby a single NCOA4 molecule (or more likely two NCOA4 molecules given the oligomerization-mediating N-terminal coiled coil domain in NCOA4) is sufficient for

targeting one 24-subunit ferritin complex for degradation. This finding is significant given the varying composition of ferritin complexes in different cell types including cell types with predominantly FTL expression (e.g., spleen and liver) (*Arosio et al., 2009*). The expectation is that even in cells predominantly expressing FTL, NCOA4 would be able to function to promote ferritinophagy. This finding also suggests that even small changes in the pool of NCOA4 would be sufficient to alter ferritinophagy levels. Further delineation of the binding determinants of the NCOA4-FTH1 complex will require structural approaches.

## Iron-dependent NCOA4 turnover as a means to control ferritinophagy

Our data reveal that HERC2 associates with NCOA4 in an iron-dependent manner, and for the first time, provide evidence that NCOA4 itself is an iron-sensing protein capable of coordinating iron via a C-terminal helical domain. While the sheer size of HERC2 precludes traditional in vitro studies, we demonstrate that HERC2 ablation or overexpression of HERC2[F3–6] (amino acids 2540–2700 containing the NCOA4 binding site) stabilizes endogenous NCOA4 under iron-replete conditions. We noted that NCOA4 remains unstable, albeit with significantly slower kinetics, upon HERC2 depletion, and pre-treatment with the proteasome inhibitor Bortezomib rescues, but does not completely restore NCOA4 protein expression following CHX treatment. Likewise, blockade of autophagy only partially stabilizes NCOA4. These data indicate that NCOA4 levels are under the control of dual autophagic and proteasomal systems, and both are operative in the setting of intermediate iron levels as observed under typical cell culture conditions. Turnover through autophagy could be linked specifically to its capture during the process of ferritinophagy, although we cannot rule out the possibility that NCOA4 also has additional cargo that may also contribute to its flux through the autophagy system. When iron is available, the iron bound pool of NCOA4 can be targeted for ubiquitylation by HERC2, leading to NCOA4 turnover through the proteasome. When iron levels are low, such as modeled here by chelation, a pool of NCOA4 is liberated from the grasp of HERC2 and may then participate in trafficking of ferritin to the lysosome. Inappropriate depletion or accumulation of FTH1 may also provide insight into the dynamics of NCOA4 turnover. While FTH1 knockout leads to a marked decrease in NCOA4 levels (*Figure 2D*), likely due to an increase in free iron thereby activating HERC2-mediated turnover, over-expression of wild type FTH1 leads to a marked accumulation of NCOA4 (*Figure 2E*, *Figure 2—figure supplement 1C*). This may be due to an over-abundance of ferritin complexes chelating free iron (as suggested by a rise in IRP2 levels, *Figure 2E*, *Figure 2—figure supplement 1C*); however, as re-expression of FTH1[R23A] in FTH1-null cells leads to only a modest increase in NCOA4 there is likely some ability of ferritin complexes to sequester NCOA4 from degradation via HERC2 and even autophagy. A model that depicts multiple modes of regulation of NCOA4 is provided in *Figure 6*. The structural determinants of NCOA4 iron binding and how iron is loaded onto NCOA4 have yet to be determined but we do note the presence of multiple conserved cysteine residues throughout NCOA4 that may play a role in iron binding.

The mechanism of NCOA4 regulation by HERC2 contrasts with the recently described role of HERC2 in degradation of the iron-sensing protein FBXL5, as HERC2-mediated FBXL5 turnover was demonstrated to occur basally, independent of FBXL5 iron binding capacity (*Moroishi et al., 2014*). Upon intracellular iron depletion, FBXL5 is rapidly destabilized via the activity of an unidentified E3 ubiquitin ligase, and loss of HERC2 function cannot restore FBXL5 expression under such conditions (*Moroishi et al., 2014*). FBXL5 loss leads to stabilization of IRP2, a ferritin translational repressor (*Salahudeen et al., 2009*; *Vashisht et al., 2009*). Together, NCOA4-mediated ferritinophagy and IRP2-mediated inhibition of ferritin translation function to mobilize intracellular iron stored within ferritin complexes and prevent further iron storage until intracellular iron levels are restored (*Figure 6*).

## NCOA4 controls erythroid development

Erythropoiesis is defined by a complex series of differentiation steps beginning with the hematopoietic stem and progenitor cell leading to the mature enucleated discoid erythrocyte replete with hemoglobin. One of the defining aspects of erythroid differentiation is the requirement for massive amounts of iron to support the synthesis of increasing amounts of hemoglobin in the late stages of erythroblast maturation (*An et al., 2014*). Indeed, of the approximately 3–5 g of iron in the human body, greater than 2 g is present as heme in hemoglobin of erythrocytes (*Pantopoulos et al., 2012*). While it is clear that iron is delivered to erythroblasts via circulating transferrin and that this iron is required for heme synthesis in

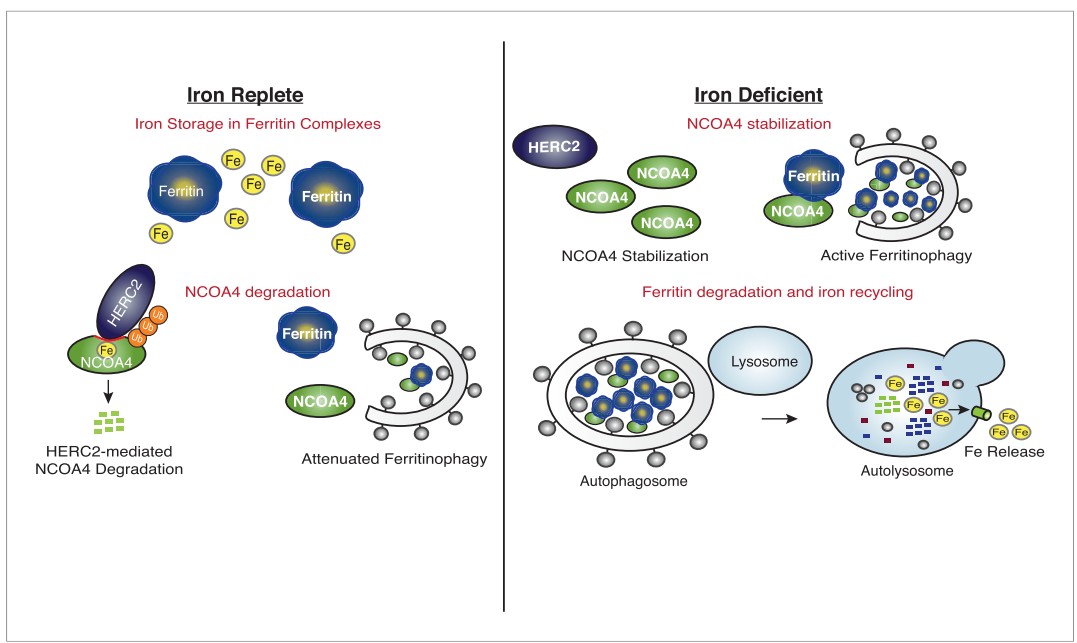

**Figure 6**. Iron Levels Regulate NCOA4-mediated ferritinophagy. A model of NCOA4 ferritinophagy regulation (see the text for details).

the mitochondria, the intracellular itinerary of iron once liberated from transferrin is unclear in the developing erythroblast. Prior studies suggest either direct delivery of endosomal iron to the mitochondria vs an intermediate storage step in ferritin followed by lysosomal degradation of ferritin with subsequent iron release (*Vaisman et al., 1997*; *Sheftel et al., 2007*). Our findings suggest that in the developing zebrafish and during hemin-induced erythroid-like K562 differentiation, the majority of iron obtained via endocytosed holo-transferrin is stored in ferritin before release via NCOA4-mediated ferritinophagy. In support of this model is a recent transcriptome analysis showing a significant upregulation of *NCOA4* mRNA into the top 25 highest expressed genes at the orthochromatic erythroblast stage of erythroid differentiation, the stage associated with massive heme and hemoglobin synthesis (*An et al., 2014*). While *FTL* mRNA remains within the top 25 mRNAs expressed throughout all terminal erythroid differentiation steps, *FTH1* mRNA peaks at the final two stages (polychromatic erythroblast and orthochromatic erythroblast). As NCOA4 specifically recognizes FTH1 and not FTL, this could suggest that during erythroid development there is a switch in the composition of ferritin complexes from predominantly FTL favoring storage of iron to a mix of FTH1-FTL. At later stages when iron is required for heme biosynthesis, NCOA4 could then more efficiently and with better kinetics target for degradation ferritin complexes with this higher FTH1-FTL ratio. The modulation of *NCOA4* transcript levels during erythroid differentiation suggests an additional layer of regulation of *NCOA4* at the transcriptional level that we do not yet understand beyond the post-translational roles of HERC2 and autophagy.

The recent results from Dowdle et al. support the role of NCOA4 in maintaining iron homeostasis in vivo as *ncoa4* knockout led to inappropriate accumulation of iron in mouse splenic macrophages (*Dowdle et al., 2014*). The authors do not specifically note a defect in erythropoiesis in their mice; however, as splenic macrophages play an important role in turnover of senescent red blood cells and a role in maintaining erythropoiesis during pathological erythropoiesis states, this suggests some defect in erythropoiesis for which splenic macrophages may be compensating (*Ramos et al., 2013*). On the other hand, the role of NCOA4 in erythropoiesis could be limited to the setting of embryonic development and stress erythropoiesis which our zebrafish and K562 model systems are best suited to evaluate. Further study of NCOA4 in vivo in specific tissue compartments, including the erythropoietic compartment, will be necessary in order to decipher the role of NCOA4 in not only erythropoiesis but also overall organismal iron metabolism.

## Materials and methods

### Mammalian cell culture

Cells were cultured in a humidified incubator at 37°C and 5% $CO_2$. 293T, U2OS, HCT116, and K562 cell lines were obtained from the American Type Culture Collection (ATCC, Manassas, VA, United States) and tested for mycoplasma contamination. The aforementioned cell types were maintained in high glucose-containing DMEM (Invitrogen, Grand Island, NY, United States) supplemented with fetal bovine serum (Hyclone, GE Life Sciences, Logan, UT, United States). K562 cells were maintained in IMDM (ATCC) supplemented with fetal bovine serum and antibiotics.

### Antibodies and chemicals

The following antibodies were used in this study: Flag M2 monoclonal (Sigma [St. Louis MO, United States] F1804-200UG; Western 1:1000), HA.11 Clone 16B12 monoclonal (Covance [Dedham, MA, United States] MMS-101P; Western 1:2000), FTH1 (Cell Signaling [Danvers, MA, United States] 3998 and 4393, Western1:1000), Ferritin (Rockland [Limerick, PA, United States] 200-401-090-0100, IF 1:400), FTL (Abnova [Taiwan, China] Ab69090; Western 1:1000), NCOA4 (Bethyl Laboratories [Montgomery, TX, United States] A302-272A; Western 1:5000 and Santa Cruz [Dallas, TX, United States] 373739; Western 1:100), NCOA4 (Bethyl A302-271A; IP 8 µg/mg protein), HERC2 (BD Transduction Labs [San Jose, CA, United States] 612366; Western 1:1000), HERC2 (Bethyl A301-905A; IP 4 µg/mg protein), IRP2 (Santa Cruz Sc-33682; Western 1:500), β-Actin (Santa Cruz, western blot 1:10,000), Hemoglobin γ (Cell Signaling 14,818, western blot: 1:1000), LAMP2 (Abcam [Cambridge, MA, United States] Ab25631, immunofluorescence 1:100). Secondary antibodies: Anti-Rabbit IgG (H + L) or Anti-Mouse HRP Conjugate (Promega [Madison, WI, United States] w4011, w4021; Western 1:7500), Alexa Fluor 488 anti-Mouse IgG (H + L) (IF 1:1000); Alexa Fluor 594 anti-Rabbit IgG (IF 1:1000). The following chemicals were used for treatment of cells as indicated: Deferoxamine (DFO, Sigma, 100 µM), Deferasirox (DFX; Selleckchem [Houston, TX, United States]; 30 µM) Ferric Ammonium Citrate (FAC, Fisher Scientific [Pittsburgh, PA, United States], 0.025–0.1 mg/ml), Bafilomycin A1 (Sigma, 50 nM), E64-d (Sigma; 10 µg/ml), PepstatinA (CalBiochem [Billerica, MA, United States]; 10 µg/ml), Bortezomib (gift from Millenium Pharmaceuticals; 1 µM) and Cycloheximide (Sigma, 100 µg/ml).

### cDNA expression constructs

Six fragments spanning the HERC2 ORF cloned in the pFLAG-CMV vector were a gift from Neils Mailand (University of Copenhagen). These fragments served as template for PCR amplification and BP Gateway cloning into pDONR223. Subsequently, LR reactions were carried out to transfer each fragment into the CMV-driven pHAGE-N-Flag-HA lentiviral vector for expression in cells Fragments 1–6, F1–F6. Smaller sub-fragments of HERC2$^{F3}$ were obtained by BP PCR and LR Gateway cloning into the pHAGE-N-Flag-HA vector. Constructs were expressed in the indicated cell lines by transient transfection using polyethylenimine (PEI). The human NCOA4α isoform (amino acids 1–614, NM_001145263.1, wild type or point mutant) lacking the stop codon was cloned into pHAGE-C-FLAG-HA, and human FTH1 (NM_002032.2, wild type or point mutant) was cloned into pHAGE-N-FLAG-HA or pHAGE-N-GFP using Gateway recombination. Gene-encoding vectors were packaged into lentiviral particles in 293T cells, and cell lines stably expressing FLAG-HA or GFP tagged proteins were generated by lentiviral transduction and puromycin selection (1 µg/ml). Point mutations in NCOA4 and FTH1 were generated using site-directed mutagenesis using KOD polymerase (Millipore, Billerica, MA, United States).

### RNAi and CRISPR-mediated genome editing

The following small interfering RNA (siRNA) were obtained from Dharmacon (Logan, UT, United States): Control non-targeting siRNA (D-001210-01-05) and ON-TARGET plus Human HERC2 (NM_004667.5); siHERC2-1: 5′-GCACAGAGUAUCACAGGUA-3′; and siHERC2-2: 5′-CGAUGAAG GUUUGGUAUUU-3′. siRNAs (20 µM) were reverse transfected using RNAiMax (Invitrogen). Lentiviral shRNA plasmid clones (pLKO.1) were obtained from the RNAi Consortium collection. shNCOA4-1: 5′-CCCAGGAAGTATTACTTAATT-3′ (TRCN0000019724); shNCOA4-2: 5′-GGCCCAGGAAGTAT TACTTAA-3′ (TRCN0000236185); and shGFP: 5′-GCAAGCTGACCCTGAAGTTCAT-3′ (Addgene [Cambridge, MA, United States] plasmid #30323) (NCOA4 accession number NM_001145263.1). Clustered Regularly Interspaced Short Palindromic Repeats (CRISPR)/Cas9-mediated genome editing

was used to generate clonal NCOA4 and FTH1 knockout HCT116 cell lines. High-quality gRNA sequences were designed using the crispr.mit.edu resource, and guides were cloned into the pX330 vector (Addgene) for simultaneous gRNA and Cas9 expression (Cong et al., 2013; Ran et al., 2013). The following gRNAs were used in our study: NCOA4 (exon 2)—GTCTTAGAAGCCGTGAGGTA and FTH1 (exon 1)—GACCATGGACAGGTAAACGT. Cells were transfected using PEI with px330 empty vector (control) or px330 vector containing gene-specific gRNA, along with our pHAGE-N-FLAG-HA-GFP vector (harboring puromycin resistance). 48 hr post-transfection, cells were treated with 1 µg/ml puromycin to select for transfected cells followed by limiting dilution plating to obtain 1 cell/well on 96-well plates. Clonal colonies were screened for NCOA4 or FTH1 protein expression by western blotting and targeting was confirmed by sequencing analysis.

## Immunological methods

For FLAG affinity purification of NCOA4-FLAG-HA (CTAP) or FLAG-HA-FTH1 (NTAP) wild type or mutant variants, HCT116 cells were harvested at ~80% confluency and lysed in 50 mM Tris–HCl (pH 7.5), 150 mM NaCl, 0.5% Nonidet P40, 1 mM DTT and protease inhibitors (Roche, Penzberg, Germany). Cleared extracts were subjected to IP with anti-FLAG M2 Magnetic Beads (Sigma M8823). Complexes were washed with lysis buffer and subjected to SDS-PAGE and immunoblot with the indicated antibodies. For FLAG affinity purification of HERC2 fragments, 293T cells expressing the indicated constructs were harvested, lysed, and subjected to anti-FLAG IP as described above. FLAG elutions were performed using (250 µg/ml) 3× FLAG peptide (LabPe L1033) and immunoblots were performed as indicated. For endogenous HERC2 and NCOA4 IPs, 293T cells were treated as indicated, harvested and lysed as noted above. Cleared extracts were subjected to IP with recombinant Protein G Agarose beads (Invitrogen 15920-010) pre-coupled with protein-specific antibodies as described previously. For an IP control, untreated extracts were incubated with Protein G Agarose coupled with Normal Rabbit IgG (Cell Signaling #2729). To examine endogenous ferritin or GFP-FTH1 localization, cells were plated on glass coverslips, treated as indicated, and fixed with 4% paraformaldehyde before immunofluorescence using anti-ferritin or LAMP2 antibodies (for endogenous staining) or GFP visualization (GFP-tagged FTH1). Images were collected using a Yokogawa (Sugar Land, TX, United States) CSU-X1 spinning disk confocal with Borealis modification on a Nikon (Melville, NY, United States) Ti-E inverted microscope using an X100 Plan ApoNA 1.4 objective lens. Images were acquired with a Hamamatsu (Middlesex, NJ, United States) ORCA-AG cooled CCD camera controlled with MetaMorph 7 software (Molecular Devices, Sunnyvale, CA, United States). Z-series optical sections were collected with a step size of 0.2 µm, using the internal Nikon Ti-E focus motor and stacked using MetaMorph to construct maximum intensity projections.

## Protein expression and purification

Myc-tagged NCOA4 protein for experiments in *Figure 1B* were produced using the 1-step human coupled in vitro translation kit (Thermo Fisher Scientific, Waltham, MA, United States) given *E. coli* expressed NCOA4α was insoluble. NCOA4 constructs for *Figure 1B* were cloned into a Thermo Scientific pT7CFE1-NMyc vector modified for Gateway cloning (NCOA4β was obtained by BP PCR Gateway cloning from a MCF7 cell line cDNA library). NCOA4-GST truncation and mutant constructs for the remainder of the paper were expressed using a dual-tagging system of N-terminal $His_6$-Sumo ($His_6$-Sumo obtained from MRC-PPU, University of Dundee, Plasmid DU40847) and C-terminal GST, utilizing both $Ni^{2+}$-IMAC and glutathione Sepharose chromatography for purification with removal of the $His_6$-Sumo moiety by Senp1 protease cleavage (MRC-PPU, University of Dundee, Plasmid DU39129). Briefly, plasmids were transformed in *E. coli* BL21 (DE3) Rosetta cells. Protein expression was induced with 1 mM IPTG for 3 hr at 37°C. Cells were collected by centrifugation, re-suspended in 25 mM Tris–HCl (pH 7.4), 10% (wt/vol) sucrose and lysed with a single freeze–thaw cycle. DNase I was added to reduce the viscosity and following centrifugation, protein was purified by $Ni^{2+}$-IMAC chromatography. Eluted proteins were subsequently purified using glutathione sepharose beads and the $His_6$-Sumo moiety was cleaved by Senp1 protease. Purified GST-fusion constructs were used for GST-pulldowns as below. FTH1-wild type and mutant, FTL, $HERC2^{2540-2700}$, $HERC2^{2553-2639}$, $HERC2^{2631-2739}$, and $NCOA4^{383-509}$ were produced as C-terminal fusions to $His_6$-Sumo. Proteins were purified by $Ni^{2+}$-IMAC chromatography followed by cleavage with Senp1 protease. The cleavage reaction was passed over a $Ni^{2+}$-IMAC 'catch' column, and cleaved protein products were collected in the flow

through fractions. Cleaved proteins were purified further with Q-sepharose, and gel filtration chromatography on either a Superose 6 column (FTH1 and FTL complexes) or a Superdex 75 column (HERC2 and NCOA4 proteins).

## GST-pulldowns

Recombinant GST fusion proteins were incubated with 10 µl of a 50% (vol/vol) slurry of glutathione Sepharose 4B (GE Healthcare, Pittsburgh, PA, United States) beads for 30 min at 4°C. For in vitro translated Myc-tagged NCOA4 proteins, anti-c-Myc agarose affinity gel (Sigma) was used. Beads were washed three times with 150 mM NaCl, 20 mM Tris (pH 7.4), 2 mM DTT, 0.5% (wt/vol) Nonidet P-40 and then mixed with purified proteins as indicated in experiments. For ferritin pulldowns, apoferritin purified from equine spleen (Sigma A3641) was added at either 2 µg (for immunoblot-based pulldowns) or 50 µg (for coomassie-based pulldowns) per pulldown. For recombinant FTH1, FTL, or HERC2 pulldowns, 50 µg of purified protein was added per pulldown. The assay mix was incubated for 30 min at 4°C, and beads were washed four times with 1 ml wash buffer. Proteins were eluted with SDS sample buffer and analyzed by 4–20% gradient SDS–PAGE followed by Colloidal Coomassie Blue stain or, for the indicated experiments with immunoblotting using FTH1 antibody. Load lane for each experiment is 5% of the input.

## ICP-MS

Aliquots of buffer only, HERC2$^{2553–2639}$, and NCOA4$^{383–509}$ mixed with 500 µl Aristar Ultra nitric acid (VWR, Radnor, PA, United States) followed by digestion for 24 hr. Samples were then diluted to 5 ml with deionized water and were analyzed using a dynamic reaction cell-inductively coupled plasma mass spectrometer (DRC-ICP-MS, Elan 6100, Perkin Elmer, Norwalk, CT, United States). Samples were analyzed by the external calibration method using seven standards with concentrations ranging from 0 to 200 parts per billion (ppb). Quality control measures included: analysis of initial calibration verification standard, continuous calibration standard, procedural blanks, and duplicate samples. Reported measurements of Fe in each sample represent the average of 5 measurements. The individual measurements are typically obtained with a relative standard deviation of approximately 2.5% or less per sample.

## K562 differentiation experiments

K562 cells were transduced with shGFP or shNCOA4 lentivirus, followed by puromycin selection to enrich for shRNA-expressing cells. To promote differentiation cells were treated with 25 µM hemin for 72 hr. Cells were pelleted and washed with PBS, followed by downstream analysis of hemoglobin expression by western blot or quantitative RT-PCR. Total RNA was extracted using TRIzol (Invitrogen) and reverse transcription was performed from 2 mg of total RNA using oligo-dT and MMLV HP reverse transcriptase (Epicentre, Madison, WI, United States), according to the manufacturer's instructions. Quantitative RT–PCR was performed with SYBR Green dye using an Mx3000PTM instrument (Stratagene, Santa Clara, CA, United States). PCR reactions were performed in triplicate and the relative amount of cDNA was calculated by the comparative $C_T$ method using the 18S ribosomal RNA sequences as a control. Primer sequences were as follows: Hemoglobin γ (HBG1): Forward: ACAAGCCTGTGGGGCAA, Reverse: GCCATGTGCCTTGACTTT; Hemoglobin α (HBA2): Forward: TCTCCTGCCGACAAGACCAA, Reverse: GCAGTGGCTTAGCTTGAAGTTG.

## Zebrafish husbandry and embryonic experiments

Zebrafish were maintained according to Institutional Animal Care and Use Committee protocols. Transgenic line globin-LCR:eGFP was described previously (Ganis et al., 2012). Embryos were processed for in situ hybridization using standard protocols (http://zfin.org/ZFIN/Methods/Thisse-Protocol.html). An antisense probe for ncoa4 was generated from a PCR template using the following primers targeting the 3′ end of the ncoa4 transcript: forward = CCTCTGGAGAGCACATGCAA, reverse = TGCATCCAGTCCACTTGTTCT. Morpholino (GeneTools, Philomath, OR, United States) knockdown was performed as previously described (Goessling et al., 2008). Morpholino sequences for ncoa4 knockdown are as follows: ncoa4 MOa (targeting the ATG start codon): 5′-CTCTCTGCCCCATAGGAGACATACT-3′, ncoa4 MOb (targeting the exon2/intron2 splice site): 5′-ATGCCAAAAACACGTCTCACCTCTC-3′. Control morpholino (GeneTools): 5′-CCTCTTACCTCAGTTACAATTTATA-3′.

Flow cytometry of whole embryos (pools of 5 embryos, > 10 biological replicates) was performed by first dissociating embryos enzymatically using Liberase 50 µg/ml (Roche) at 37°C for 2 hr followed by mechanical dissociation by pipetting. The dissociation was stopped by washing with PBS and the cells were filtered (30 µm) prior to analysis. FACS analysis was performed using a FACS Canto II (BD Biosciences, San Jose, CA, United States), and SYTOX Red staining was used to gate out dead cells. Flow cytometry data were analyzed using FACSDiva (BD Biosciences).

## Acknowledgements

We thank Paul Schmidt and Mark Fleming for helpful discussions regarding erythroid differentiation experiments, Neils Mailand for providing expression constructs spanning the HERC2 CDS and the Nikon Imaging Center (Harvard Medical School) for use of the Nikon Ti spinning disk confocal microscope, Metamorph software, and imaging support. We thank the Harvard School of Public Health Trace Metals Laboratory for assistance with ICP-MS. This work was supported by NIH grants GM070565 to J.W.H. and GM095567 to JWH and ACK, and National Cancer Institute Grants R01CA157490, R01CA188048, ACS Research Scholar Grant RSG-13-298-01-TBG, and Lustgarten Foundation, to ACK. JDM is supported by an American Society of Radiation Oncology Junior Faculty Career Research Training, a KL2/Catalyst Medical Research Investigator Training award (TR001100), and a Burroughs Wellcome Fund Career Award for Medical Scientists. LPV is supported by a Damon Runyon Post-Doctoral Fellowship. SN is supported by NIH NIDDK K08 DK105326-01 and a Burroughs Wellcome Fund Career Award for Medical Scientists.

## Additional information

### Competing interests

ACK: reports receiving speakers bureau honoraria from Agios and the US Oncology Network, and is a consultant for Astellas Pharma, FORMA Therapeutics, and Gilead. JWH: is a consultant for Millennium: the Takada Oncology Company and Biogen-Idec. The other authors declare that no competing interests exist.

### Funding

| Funder | Grant reference | Author |
| --- | --- | --- |
| National Institutes of Health (NIH) | GM095567 | Alec C Kimmelman, J Wade Harper |
| National Cancer Institute (NCI) | R01CA157490 | Alec C Kimmelman |
| American Cancer Society (ACS) | RSG-13-298-01-TBG | Alec C Kimmelman |
| Lustgarten Foundation | | Alec C Kimmelman |
| American Society for Radiation Oncology (ASTRO) | JF2013-2 | Joseph D Mancias |
| Burroughs Wellcome Fund (BWF) | Career Award for Medical Scientists | Joseph D Mancias, Sahar Nissim |
| Damon Runyon Cancer Research Foundation (Damon Runyon) | | Laura Pontano Vaites |
| National Cancer Institute (NCI) | R01CA188048 | Alec C Kimmelman |
| National Institutes of Health (NIH) | GM070565 | J Wade Harper |
| National Institutes of Health (NIH) | KL2 TR001100 | Joseph D Mancias |
| National Institutes of Health (NIH) | DK105326-01 | Sahar Nissim |

The funders had no role in study design, data collection and interpretation, or the decision to submit the work for publication.

## Author contributions

JDM, LPV, SN, DEB, AJK, XW, YL, WG, ACK, JWH, JDM, LPV, ACK, and JWH conceived the experiments. JDM and LPV performed all experiments except for zebrafish studies in Figure 5. SN, AJK, and WG performed zebrafish related experiments in Figure 5. DEB, XW, and YL provided technical support for mutagenesis, sequencing of CRISPR clones, cell line generation, and quantitative RT-PCR. JDM, LPV, ACK, and JWH wrote the manuscript, and all authors edited the manuscript

## Ethics

Animal experimentation: Zebrafish were maintained according to Institutional Animal Care and Use Committee protocol #04626 of the Harvard Medical Area Standing Committee on Animals.

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
