## [Decision Letter]

Thank you for submitting your work entitled “Ferritinophagy via NCOA4 is required for erythropoiesis and is regulated by an iron dependent HERC2-mediated proteolysis” for peer review at *eLife*. Your submission has been favorably evaluated by Vivek Malhotra (Senior Editor) and three reviewers, one of whom is a member of our Board of Reviewing Editors.

The following individuals responsible for the peer review of your submission have agreed to reveal their identity: Ivan Dikic (Reviewing Editor) and Andrew Thorburn (peer reviewer).

The reviewers have discussed the reviews with one another and the Reviewing editor has drafted this decision to help you prepare a revised submission.

Mancias and colleagues follow up their recent finding that NCOA4 mediates ferritinophagy by reporting how FTH1 interacts with NCOA4 and provide new insights into NCOA4 regulation by their discovery that in addition to being degraded in the lysosome by autophagy (as one might expect given their previous identification of this protein as a selective autophagy receptor). In this process, NCOA4 captures ferritin with bound iron and directs it into the autophagy pathway where it is degraded, liberating bound iron. In the present manuscript, they make some key discoveries. First, the C-terminus of NCOA4 binds directly to FTH1 through a critical surface-exposed R residue. Second, not only autophagy, but also the ubiquitin pathway controls NCOA4 levels via iron binding and HERC2-directed degradation. Third, NCOA4 is important for erythroid differentiation in two different models, establishing the physiological context where this process is important.

The reviewers were impressed with the high quality of the data presented, but they felt that one potential improvement to the manuscript may come from functional analysis of the rescue phenotypes with wt and the point mutants in NCOA4^I489A/W497A^ and/or the FTH1^R23A^ mutant that disrupt interaction and ferritinophagy in the K562 cell culture model and possibly also in vivo in the fish model. However, the reviewers agreed that this is not essential for the acceptance of the manuscript, but if the authors have such experiments already underway they are invited to present them in a revised manuscript.

---

## [Author Response]

*The reviewers were impressed with the high quality of the data presented, but they felt that one potential improvement to the manuscript may come from functional analysis of the rescue phenotypes with wt and the point mutants in NCOA4*^*I489A/W497A*^
*and/or the FTH1*^*R23A*^
*mutant that disrupt interaction and ferritinophagy in the K562 cell culture model and possibly also* in vivo *in the fish model. However, the reviewers agreed that this is not essential for the acceptance of the manuscript, but if the authors have such experiments already underway they are invited to present them in a revised manuscript*.

With regards to the optional rescue experiments suggested in the review, we do not currently have these experiments under way and would therefore request resubmission of the manuscript in its current state. While this additional optional experiment using the K562 system would be of interest, we feel that the current data of rescue with wild-type vs mutant NCOA4 or FTH1 in the HCT116 knockout cell lines provided clear evidence that mutant NCOA4 or FTH1 cannot facilitate ferritinophagy. Therefore, these additional experiments would not add significantly to the conclusions of the paper. We previously sought to address wild-type vs mutant NCOA4 function in K562 cells by deleting NCOA4 as we did in the HCT116 system followed by reconstitution with mutant NCOA4; however, we were unable to obtain viable NCOA4 knockout clones. Attempts to knockout FTH1 in K562s were similarly unsuccessful. This supports our data showing the importance of iron/ferritinophagy in this cell type as it is unable to tolerate complete knockout and the pressure of repopulating from a single cell clone. As such we used a previously validated RNAi approach to modulate NCOA4 expression. In addition to the K562 experiments, we used independently designed RNAi to zebrafish NCOA4 and obtained similar results, strengthening our argument with respect to the importance of NCOA4 in erythropoiesis.